# ViPE: Visualise Pretty-much Everything

**Hassan Shahmohammadi**     **Adhiraj Ghosh**     **Hendrik P. A. Lensch**
University of Tübingen
`hassan.shahmohammadi@uni-tuebingen.de`

## Abstract

Figurative and non-literal expressions are profoundly integrated in human communication. Visualising such expressions allow us to convey our creative thoughts, and evoke nuanced emotions. Recent text-to-image models like Stable Diffusion, on the other hand, struggle to depict non-literal expressions. Recent works primarily deal with this issue by compiling humanly annotated datasets on a small scale, which not only demands specialised expertise but also proves highly inefficient. To address this issue, we introduce ViPE: Visualise Pretty-much Everything. ViPE offers a series of lightweight and robust language models that have been trained on a large-scale set of lyrics with noisy visual descriptions that represent their implicit meaning. The synthetic visual descriptions are generated by GPT3.5 relying on neither human annotations nor images. ViPE effectively expresses any arbitrary piece of text into a visualisable description, enabling meaningful and high-quality image generation. We provide compelling evidence that ViPE is more robust than GPT3.5 in synthesising visual elaborations. ViPE also exhibits an understanding of figurative expressions comparable to human experts, providing a powerful and open-source backbone to many downstream applications such as music video and caption generation.

## 1 Introduction

*"Language is the dress of thought."* - Samuel Johnson

How do humans comprehend such a metaphorical phrase? Conceptual metaphors play a significant role in shaping our language, enabling us to relate concrete experiences and emotions with abstract concepts (Lakoff and Johnson, 2008). They serve as powerful tools for conveying intricate ideas, highlighting emotions, and adding a sense of humour to our statements. In addition, visualising

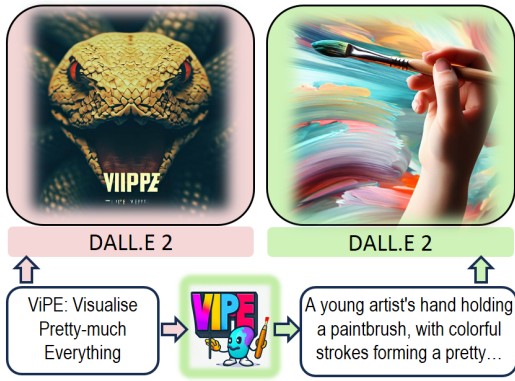

Figure 1: Given any arbitrary text, ViPE composes multiple meaningful textual illustrations, thereby assisting state-of-the-art text-to-image models in effectively conveying the intended message via visual symbols.

metaphorical phrases and abstract concepts allows us to express our creative ideas (Schwering et al., 2009). In advertising, they frequently serve as persuasive tools to evoke positive attitudes (Phillips and McQuarrie, 2004; McQuarrie and Mick, 1999; Jahameh and Zibin, 2023). While humans effortlessly interpret images with metaphorical content (Yosef et al., 2023), state-of-the-art text-to-image models such as DALL.E 2 (Ramesh et al., 2022) and Stable Diffusion (Rombach et al., 2022) still struggle to synthesise meaningful images for such abstract and figurative expressions (Kleinlein et al., 2022; Chakrabarty et al., 2023; Akula et al., 2023).

Recent efforts in addressing this challenge have mostly focused on constructing datasets for figurative language, such as metaphors, similes, and idioms (Chakrabarty et al., 2023; Yosef et al., 2023; Akula et al., 2023). However, these datasets are often small in size and require expert knowledge for expansion. Moreover, despite the benefits of these datasets, the fundamental issue of text-to-image models remains unresolved. To address these limitations, we present ViPE: Visualise Pretty-much Everything. ViPE eliminates the need for human annotations or images with metaphorical contents,

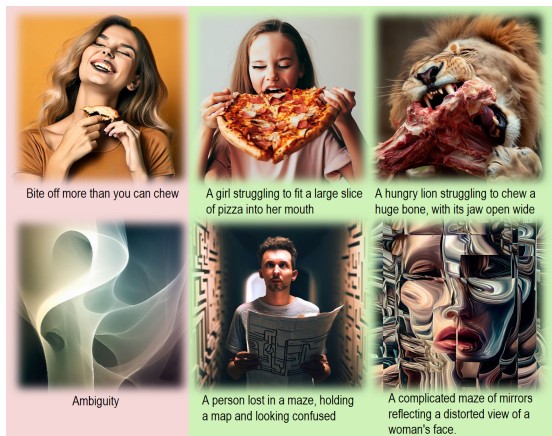

Figure 2: ViPE enhances the visualisation of figurative language and abstract concepts for text-to-image models. DALL.E 2 (left) struggles to depict such phrases. ViPE successfully captures the implicit meanings and communicates them through visual symbols.

yet effectively assists text-to-image models in visualising figurative and abstract phrases, and even arbitrary textual input. The core idea behind our approach is to unfold the implicit meaning through a new textual description (elaboration) containing visual symbols. Following (Chakrabarty et al., 2023), we use the term *Visual Elaboration* to refer to the visualisable textual description of a piece of text with figurative content. As illustrated in Figure 1, ViPE transforms the input into a detailed image caption while preserving the intended meaning. Therefore, it facilitates the visualisation of figurative language. Building ViPE involves three main stages. **(1) A Large Scale Lyric dataset:** we compile a large-scale collection of lyrics ($\approx$ 10M lines) as a rich source of figurative language. **(2) Synthetic Visual Elaborations:** we construct a supervised dataset we call **LyricCanvas**, by employing a Large Language Model(LLM) to generate noisy visual elaborations for all the lyrics. **(3) Knowledge Distillation:** we conduct knowledge distillation to build a robust model by fine-tuning a set of lightweight language models on LyricCanvas.

ViPE, our approach, addresses the limitations found in previous works by leveraging two key findings. The first finding is that lyrics serve as a rich repository of knowledge, embodying a wide spectrum of figurative language, including metaphors, similes, idioms, and beyond (Chakrabarty et al., 2021; Swarniti, 2022; Astina et al., 2021). The second finding stems from the observation that the task of ViPE is akin to style transfer using machine translation (MT) (Zhang et al., 2018; Shen

et al., 2017; Li et al., 2022b), which often benefits from large amounts of data (Hassan et al.; Edunov et al., 2018; Britz et al., 2017), including noisy data (Rolnick et al., 2017; Vaibhav et al., 2019; Karpukhin et al., 2019). Therefore, we propose to create a large-scale dataset, the LyricCanvas dataset, from publicly available lyrics with automated but potentially noisy visual elaborations generated by an LLM, GPT3.5[1], instructed via prompting. Subsequently, we build ViPE by fine-tuning two lightweight language models, GPT2-Small, and GPT2-Medium (Radford et al., 2019) on the LyricCanvas dataset. We will show that ViPE, despite its size (S: 117M and M: 345M parameters), is more robust than GPT3.5 with 175B parameters in synthesising zero-shot visual elaborations. Figure 2 demonstrates two challenging examples for DALL.E 2, highlighting the improvement depictions based on ViPE.

Overall, our contributions are the following. **1.** We release a robust and powerful model tailored to assist all text-to-image models in visualising non-literal expressions. **2.** We introduce the largest dataset available for generating visual elaborations, which we refer to as LyricCanvas. With approximately 10 million samples, LyricCanvas proves to be adequate, unlike existing datasets, for fine-tuning powerful language models like GPT2. Moreover, we provide our scraper framework, allowing researchers to acquire the exact training inputs at no additional cost. **3.** We eliminate the expensive and time-consuming involvement of human expert annotations for abstract and figurative visualisations. **4.** We show that ViPE's generation is highly robust and is competitive with human experts.

ViPE's powerful zero-shot capability paves the way for its usage in downstream applications such as synthetic caption generation from keywords, abstract art visualisations, and music video generations. The source code, pre-trained ViPE, and the LyricCanvas dataset are available at [2].

## 2 Related Works

### 2.1 Text-to-Image Generation

Text-to-image synthesis has made significant progress in recent years, with diffusion-based models surpassing previous approaches such as Variational Autoencoders (VAE) (Razavi et al.,

---

[1]https://platform.openai.com/docs/models/gpt-3-5

[2]https://github.com/Hazel1994/ViPE

2019) and Generative Adversarial Networks (GANs) (Bao et al., 2017). Prominent text-to-image diffusion models include DALL.E 2 (Ramesh et al., 2022), Stable Diffusion (Rombach et al., 2022), MidJourney[3] and Craiyon[4]. Recent works have explored the integration of LLMs into these models. For instance, Opal (Liu et al., 2022c) enables structured search for visual concepts, Generative Disco (Liu et al., 2023a) facilitates text-to-video generation for music visualisation, and Reel-Framer (Wang et al., 2023) aids in transforming written news stories into engaging video narratives for journalists. Nonetheless, despite their success at generating creative imagery, they still struggle to visualise figurative language effectively (Kleinlein et al., 2022; Chakrabarty et al., 2023; Akula et al., 2023). Furthermore, research by Chakrabarty et al. (2023); Akula et al. (2023) reveals that DALL·E 2 outperforms Stable Diffusion in representing figurative language. DALL·E 2 has 3.5 billion parameters, over three times that of Stable Diffusion, and incorporates textual prompts directly to establish relevance between generated images and the provided text. In contrast, Stable Diffusion uses textual prompts through cross-attention during diffusion without explicit conditioning. Our approach, ViPE, enhances the visualization of figurative and non-literal expressions in any text-to-image model as a lightweight assistant.

## 2.2 Figurative Language Visualisation

There has been extensive research on textual figurative language such as metaphor generation (Yu and Wan, 2019; Chakrabarty et al., 2020; Terai and Nakagawa, 2010), idiom generation and paraphrasing (Liu and Hwa, 2016; Zhou et al., 2021), and simile recognition and interpretation (Zeng et al., 2020; He et al., 2022a). Visualising figurative language, on the other hand, has received less attention. Existing approaches primarily revolved around constructing datasets with images and annotations for metaphors, similes, and idioms (Chakrabarty et al., 2023; Yosef et al., 2023; Akula et al., 2023; Zhang et al., 2021). However, these datasets are small and rely on expert knowledge. For example, Chakrabarty et al. (2023) generated visual descriptions and synthetic images for 1,540 linguistic metaphors. Yosef et al. (2023) compiled a dataset of less than 3,000 figurative expressions

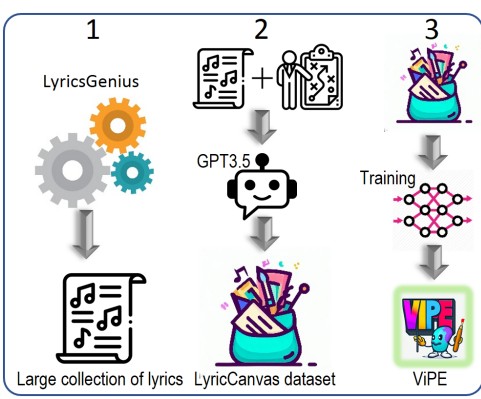

Figure 3: Building ViPE involves three main stages. 1. Constructing a large-scale dataset of lyrics. 2. Building a supervised dataset (LyricCanvas) by synthesising noisy visual elaborations using an LLM based on human instructions. 3. Training a robust and lightweight model through symbolic knowledge distillation.

with ground truth images through human annotations. Akula et al. (2023) collected 5,061 metaphorical advertisement images with a simple annotation format of "__ is as __ as __" (e.g., "this pencil is as red as a firetruck"). Zhang et al. (2021) introduced a multimodal metaphor dataset with around 10,000 samples[5]. Liu et al. (2022a) presented FigMemes, a dataset with 5,000 samples for figurative language in politically-opinionated memes.

Despite the benefits of such datasets, they do not provide a fully automated process in figurative language visualisation. We, for the first time, present a lightweight and robust model tailored for assisting text-to-image models in visualising figurative language. Our model is not only robust and open source but also requires neither human annotations nor additional images.

## 3 ViPE

We present ViPE, a set of robust and lightweight language models designed to generate visual elaborations from arbitrary text input. The development of ViPE comprises three stages, illustrated in Figure 3. **Firstly**, we perform data collection by scraping and preprocessing an extensive collection of lyrics ($\approx$ 10M lines) sourced from Genius[6]. **Secondly**, we utilise a large language model (LLM) to generate noisy visual elaborations for the lyrics by appropriate prompt design. **Finally**, the paired data of lyrics and generated visual elaborations are used to train lightweight language models. They

---

[3]https://www.midjourney.com/
[4]https://www.craiyon.com

[5]As far as we know, this dataset is not publicly available.
[6]https://genius.com/

are fine-tuned using a causal language modeling objective tailored specifically for visual elaborations. The primary goal is to generate detailed textual descriptions of visual scenes (visual elaborations) to convey the intended meaning of the rich figurative phrases in lyrics. The generated elaboration can then be passed as an input prompt to any text-to-image synthesiser to visualise the original input.

### 3.1 Data Collection

Numerous sources have been explored to capture figurative expressions (Chakrabarty et al., 2022; Liu et al., 2022b; Bizzoni and Lappin, 2018). Nonetheless, they often suffer from limitations in scale or cost. To overcome this challenge, we propose using publicly available lyrics to build a robust model. Given that the musiXmatch dataset (Bertin-Mahieux et al., 2011) is restricted to bag-of-words representations of lyrics with a maximum of only 5k unique words, the efficient integration of such datasets with modern language models becomes a non-trivial task. Therefore, we opt for scraping all the English lyrics from the Genius platform using the LyricsGenius API[7]. Subsequently, we apply a pre-processing pipeline to obtain a collection of high-quality lyrics. Our pipeline mainly includes the following filters: **Diversity:** Lyrics containing less than 15 lines with fewer than 4 unique words per song were discarded. **Length Limit:** Lines with less than 2 unique words or exceeding 20 words in total were excluded from the dataset to maintain a balanced and concise text corpus. **Size Limit:** We only used the top 50 songs from each artist sorted based on popularity to obtain a manageable dataset. The resulting dataset, referred to as the LyricCanvas dataset, comprises $\approx 10$ million lines of lyrics extracted from over 250k songs, by approximately 5.5k different artists. While we are unable to release the lyrics themselves due to copyright policies, we will make available the generated visual elaborations and the scraper and filter framework that can be employed to rebuild the LyricCanvas dataset at no additional cost.

### 3.2 Generating Initial Visual Elaborations

We propose generating synthetic visual elaborations using an LLM. Synthetic data produced by LLMs (Thoppilan et al., 2022; Brown et al., 2020; Liu et al., 2023b) offer substantial benefits

and demonstrate competitive, and in certain instances, superior performance compared to human-annotated data (He et al., 2022b; Wang et al., 2021a,b; Hu et al., 2022). A contemporary work is Chakrabarty et al. (2023), which introduces the HAIVMe dataset. There, visual elaborations are generated for 1,540 linguistic metaphors using an LLM which are subsequently refined by human experts. We use their dataset to evaluate the robustness of our model in Section 4.

In our pipeline, we instruct GPT3.5 [8], denoted as $h_T(.)$, through prompting to generate visual elaborations for a given set of lyrics. More specifically, for $(s, l, v) \in \mathcal{D}$, let $s$ be the System Role (a prefix prompt) and $l$, the set of lyrics lines corresponding to a single song in the dataset $\mathcal{D}$, we generate synthetic visual elaborations for all lines ($l_i \in l$) by conditioning the GPT3.5 model on both $s$ and $l$, as $v_i = h_T(l_i|s, l)$. Providing the surrounding lines $l$ as prior helps the model understand the theme of the song better and generate suitable visual elaborations accordingly. Our System Role contains the exact instructions to convert each line of lyrics to a meaningful visual description. The system role encompasses a total of 640 words and includes 11 distinct guidelines. For the complete system role, please refer to Appendix A.

Below, we summarise the key points covered. **Semantic Proximity:** The generated description should accurately convey the intended meaning expressed in the given line. **Visual Perceptibility:** The generated elaborations should be easily visualised. **Appropriateness:** Some lyrics contain inappropriate content, so generated output should not explicitly describe such content[9]. **Diversity:** The system is encouraged to utilise various adjectives and non-human subjects that help generate detailed and diverse images. For instance, the input line *money could be dangerous* yields *A dragon with evil eyes is lying on a pile of shiny gold*. **Emotion:** The system should further take into account the emotional tone of the context when translating lyrics into visual elaborations. This approach promotes diverse interpretations of abstract concepts.

---

[7]https://lyricsgenius.readthedocs.io/en/master/

[8]The exact version is GPT3.5 Turbo, we use GPT3.5 for simplicity

[9]We automatically discarded those lyrics that were not processed by the system due to inappropriate content.

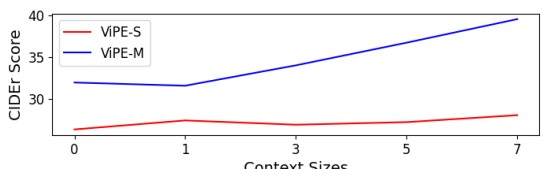

Figure 4: ViPE-Medium (ViPE-M) and ViPE-Small (ViPE-S) achieve higher CiDEr scores on the validation set of LyricCanvas with longer context.

### 3.3 Training ViPE – Generating Visual Elaboration Through Text

Training ViPE involves training a lightweight student model $h_S$ using the LyricCanvas dataset $\mathcal{D}$ with noisy labels generated by the teacher model $h_T$. In contrast to conventional knowledge distillation methods (Hahn and Choi, 2019; Hinton et al., 2015; Ba and Caruana, 2014; Chen et al., 2020), where the student is trained to predict soft labels from the teacher, we adopt an indirect approach where knowledge is transferred through discrete textual symbols. This approach, known as symbolic knowledge transfer (West et al., 2022), has been shown effective for a wide range of NLP tasks (Tang et al., 2019; West et al., 2022). In our approach, the student model $h_S$ is trained on a sequence-to-sequence task (Sutskever et al., 2014). More specifically, given a line of lyrics represented as $l_i = \{l_i^1, l_i^2, \ldots, l_i^n\}$, comprising $n$ words and its corresponding noisy visual elaboration $v_i = \{v_i^1, v_i^2, \ldots, v_i^m\}$, comprising $m$ words, our objective is to learn the conditional likelihood:

$$P(v_i|c^t) = \prod_{j=1}^{m} P(v_i^j|v_i^1, \ldots, v_i^{j-1}, c^t) \quad (1)$$

Where $c^t$ denotes the context prior, consisting of $t$ preceding lines (if available in the corresponding lyrics) relative to $l_i$ as a unified sequence. The context is prepended as a prefix to the visual elaboration $v_i$. In practice, we experiment with various context sizes. We start with a context size of zero (no context), followed by sizes of one, three, five, and seven, which follows $c^t = \{l_i, l_{i-1} \ldots l_{i-t}\}$, where $i$ and $t \in \{0, 1, 3, 5, 7\}$ correspond to the instance of a lyrics line in a song and the context length. As shown in Figure 4, by extending the context size, we provide more information to the model, thereby facilitating the generation of $v_i$ that better fits the entire lyrics.

The student $h_S$ is trained to learn the parameters $\theta$ to estimate the conditional likelihood $P_\theta(v|c^t)$

| Model | Zero-shot | Tuned (L) | Tuned (XL) |
|-------|-----------|-----------|------------|
| | Validation | | |
| GPT2 | 54.57 | 57.13 | 64.00 |
| ViPE-S | **58.50** | **61.42** | **67.28** |
| | Test | | |
| GPT2* | 53.93 | 54.80 | 62.65 |
| ViPE-S | **54.89** | **59.60** | **66.40** |

Table 1: Zero-shot and fine-tuned evaluation results using Fig-QA (Liu et al., 2022b). L and XL denote the large and X-large variations of the dataset. Our model, ViPE-S, demonstrates enhanced comprehension of figurative language compared to the standard pre-trained model. GPT2* results are from (Liu et al., 2022b)

using the standard cross entropy loss: $L_{\mathrm{xe}}(\theta) = -\log \sum_{v,c^t \in B} P_\theta(v|c^t)$, where $v$ and $c^t$ denote an instance of visual elaboration and its corresponding context in the mini-batch $B$ accordingly. We employ two versions of pre-trained GPT2 (Radford et al., 2019) as the student network $h_S$, GPT2-Small (ViPE-S) and GPT2-Medium (ViPE-M). Despite the small size of the employed GPT2 models (117M and 345M parameters), their ability to interpret the prompts has been shown very effective on both text generation (See et al., 2019) and cross-modality alignments (Nukrai et al., 2022). Furthermore, since we only condition the model on the lyrics line ($c^t$), the loss is computed only for tokens that correspond to the visual elaborations, ensuring that ViPE generates visual descriptions without generating original lyrics.

## 4 Evaluation

Assessing figurative language visualisation is a complex task due to its highly subjective nature (Figure 2). Moreover, existing evaluation procedures differ, ranging from visual entailment Chakrabarty et al. (2023), image recognition Yosef et al. (2023), and retrieval and localization Akula et al. (2023). Therefore, To fully assess the robustness of ViPE, we propose end-to-end human evaluation and various automated metrics at different levels of granularity.

### 4.1 Intrinsic Evaluation

In this section, we evaluate the general figurative language understanding of ViPE using the Fig-QA dataset (Liu et al., 2022b). It contains $\approx$ 12k figurative phrases with correct and incorrect interpretations in the Winograd style (Levesque et al., 2012). For instance, the figurative sentence *Her word had*

*the strength of a wine glass.* is paired with both *Her promises can be believed* and *Her promises cannot be trusted.* as two distinct samples. This benchmark is suitable for our purpose given that it covers various themes, including common-sense object knowledge, visual metaphor, common-sense social understanding, and cultural metaphors. We employed their evaluation framework for GPT2 and evaluated the small version of ViPE (ViPE-S) trained with the context size of one. Shown in Table 1, we compare the results of ViPE with that of GPT2 reported by Liu et al. (2022b) in both zero-shot and fine-tuned cases. The results validate the superiority of ViPE over pre-trained GPT2 in both zero-shot and fine-tuned scenarios, highlighting its advanced understanding of figurative language.

Next, we evaluate ViPE on fine-grained categories in the Fig-QA dataset (Liu et al., 2022b). As shown in Figure 7, ViPE demonstrates a comprehensive understanding of all categories in both zero-shot and fine-tuned settings. Notably, the enhancement is more prominent in the visual categories, aligning with our goal of generating visualisable descriptions for figurative language.

## 4.2 Extrinsic Evaluation

**Image-text Retrieval:** For thorough end-to-end evaluation, we conduct image-to-text and text-to-image retrieval on the HAIVMet dataset (Chakrabarty et al., 2023). HAIVMet contains 1,540 linguistic metaphors and corresponding visual elaborations reviewed by experts. We created pairs of metaphors and visual elaborations, as well as visual elaborations and images, for HAIVMet, ViPE-M trained with the context size of 7, and GPT3.5. Since HAIVMet has ground truth visual elaborations, we only generated 10 images per elaboration using Stable Diffusion (Rombach et al., 2022). For ViPE-M and GPT3.5, we generated deterministic visual elaborations for the same metaphors and then generated 10 images for each elaboration. Although the authors of HAIVMet (Chakrabarty et al., 2023) used DALL·E 2 (Ramesh et al., 2022) to generate images, we opt for a transparent and reproducible approach by utilising Stable Diffusion.

After compiling three datasets from HAIVMet, ViPE, and GPT3.5, we utilised the fine-tuned version of BLIP (Li et al., 2022a) on COCO (Lin et al., 2014) retrieval. BLIP excels in vision-language benchmarks due to the effective use of a multi-

|  | Human Experts | | GPT-3.5 | | ViPE | |
|---|---|---|---|---|---|---|
|  | TR | IR | TR | IR | TR | IR |
| Metaphor$_{zs}$ | 27.8 | **42.8** | 28.7 | 35.5 | **32.1** | 41.3 |
| Metaphor$_{ft}$ | 36.4 | **49.4** | 40.0 | 37.3 | **47.1** | 46.6 |
| Caption$_{zs}$ | 63.4 | 77.2 | 52.9 | 66.3 | **65.8** | **79.8** |
| Caption$_{ft}$ | 46.2 | 75.7 | 85.4 | 90.3 | **87.2** | **94.7** |

Table 2: A comparative report on Image-metaphor and image-caption retrieval using corpora generated by GPT-3.5, ViPE, and human experts (HAIVMet dataset) in zero-shot ($zs$) and fine-tuned ($ft$) settings. TR and IR denote the mean image-to-text and text-to-image retrieval scores respectively. ViPE outperforms GPT3.5 and shows competitive understanding to human experts.

modal encoder-decoder mixture model, making it suitable for retrieval evaluation. We used BLIP in both zero-shot and fine-tuned settings. In zero-shot, the entire retrieval dataset is used for testing, while in fine-tuned, 90% of the data is used for fine-tuning, leaving 10% for evaluation.

We report the mean recall across the top-1, top-5, and top-10 retrieval scores in Table 2. ViPE outperforms GPT-3.5 and human experts (HAIVMet) in image-metaphor retrieval (referred to as TR in the table). However, while outperforming GPT3.5, ViPE slightly lags behind humans in retrieving metaphors from images. One potential reason might be that human experts tend to be very specific in describing metaphorical images (Chakrabarty et al., 2023), creating a more discrete feature space, making it easier for BLIP to interpret. Additionally, we conduct the same evaluation on pairs of images and visual elaborations (instead of metaphors) to assess the alignment between the elaborations and corresponding images, similar to image-caption retrieval. Shown in the lower part of Table 2, ViPE outperforms both GPT3.5 and humans in both zero-shot and fine-tuned cases. An interesting finding is that GPT3.5 while showing poor performance on end-to-end evaluation, shows superior performance to humans on image-caption retrieval. This suggests that GPT3.5 prioritizes the visualisability of generated elaborations without establishing a strong connection with the metaphors. In contrast, ViPE exhibits comparable or in some cases even superior end-to-end evaluation of image-metaphor compared to humans, while also generating more detailed and concrete visual elaborations, as evidenced by the high image-caption retrieval scores.

**Emotion Visualisation:** Emotions are deeply grounded in the human visual system (Kragel et al., 2019) and computational models effectively predict emotional categories from images in various stud-

|                     | ViPE-M | GPT-3.5 |
|---------------------|--------|---------|
| Semantic Proximity  | **60.00** | 54.10 |
| Visual Perceptibility | **25.11** | 22.70 |

Table 3: A comparative analysis of ViPE-Medium and GPT3.5 in converting emotionally charged tweets into visual elaborations. ViPE is superior in generating image descriptions, demonstrating higher visual perceptibility, and preserving tweet semantics more effectively.

ies (Rao et al., 2020; Zhao et al., 2022; You et al., 2016; Achlioptas et al., 2021). We, therefore, leverage the Emotion dataset (Saravia et al., 2018) for our purpose. It is a classification dataset comprising 20k samples from Twitter messages with six basic emotions. The difficulty of visualising tweets and the plausibility of emotion detection from images puts it in line with our objective. In particular, Let $\mathcal{D}_e = \{t_i, l_i\}_1^{|D_e|}$ represent the Emotion dataset consisting of tweets $t_i$ and their corresponding labels $l_i$. Visual elaborations are generated deterministically for all tweets ($t_i \in \mathcal{D}_e$), resulting in the new dataset $\mathcal{D}_v = \{v_i, l_i\}_1^{|D_v|}$, where $v_i$ denotes the $i$th visual elaboration. This is carried out for both ViPE-M and GPT3.5, using the same System Role applied to create the LyricCanvas dataset. Subsequently, we fine-tune a pre-trained BERT model (*BERT-base-uncased*) for classification (Devlin et al., 2018) on $\mathcal{D}_v$ and evaluate the robustness of ViPE and GPT3.5 using two metrics: **Semantic Proximity (SP)** measures how well the generated visual elaboration $v_i$ represents the meaning of the tweet $t_i$, determined by the final classification accuracy on $\mathcal{D}_v$. **Visual Perceptibility (VP)** assesses the visualisability of the visual elaboration $v_i$ by computing the cosine similarity between the CLIP embeddings of $v_i$ and its corresponding generated image $I_i$ by Stable Diffusion.

The results are presented in Table 3. ViPE demonstrates superior performance in generating image descriptions, indicated by higher visual perceptibility scores. It also effectively preserves the semantic content of the tweets, as evidenced by the semantic proximity metric. Overall, our findings lend support to the efficacy of symbolic knowledge distillation (see Section 3.3) from large-scale noisy data, as demonstrated by ViPE's superior zero-shot capabilities in generating visual elaborations.

**Fine-grained Emotions:** Figure 5 compares ViPE-M and GPT3.5 in fine-grained emotion classification using the Emotion dataset. GPT3.5 leans

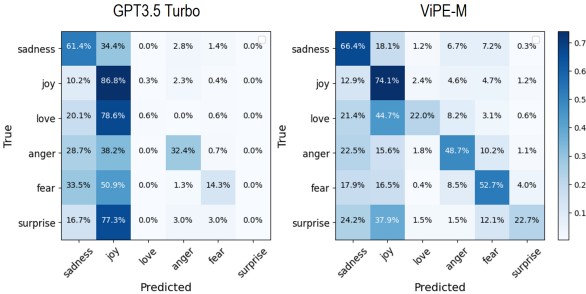

Figure 5: A comparative analysis between ViPE-M and GPT3.5 on generating visual elaboration from emotionally charged messages. ViPE-M demonstrates superior generalization performance and effectively mitigates bias towards the most dominant class ("Joy").

towards generating positive and joyful sentences, potentially influenced by positive reinforcement in its training. In contrast, ViPE-M demonstrates more precise performance and successfully mitigates bias towards the dominant class. For example, GPT3.5 shows a 77.3% confusion rate between *surprise* and *joy*, whereas ViPE reduces this bias to 37.9%. Additionally, certain emotions are challenging to distinguish solely from visual elaborations. For instance, the text *I feel that the packaging is really lovely and the product itself just does everything you ask* is labeled as *love*, but ViPE's visual elaboration of *a woman holding a beautifully wrapped gift box, smiling with excitement* is confused with *joy*.

**Safety and Appropriateness:** Even though ViPE has been fine-tuned on data generated by GPT3.5 with filtering which incorporates certain measures to mitigate inappropriate content, it is built upon the GPT-2 model which is prone to generating inappropriate content. Hence, to measure the appropriateness of ViPE's output, we conducted the following experiment. Using the Alt-profanity-check framework[10], we first measured the profanity score (inappropriate/offensive language) of the lyrics in the valuation set (around 1M line of lyrics) and distributed them over five intervals. We then measured the profanity scores of the generated visual elaborations in each interval from GPT3.5 and ViPE. In addition, we prompted the pre-trained GPT2 model with the lyrics and generated new text (not necessarily a visual elaboration). Subsequently, we measured the profanity score for the GPT2's output. Demonstrated in Figure 6, GPT2-M's scores

---

[10]https://pypi.org/project/alt-profanity-check/

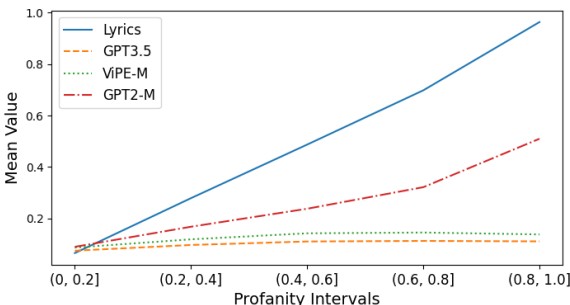

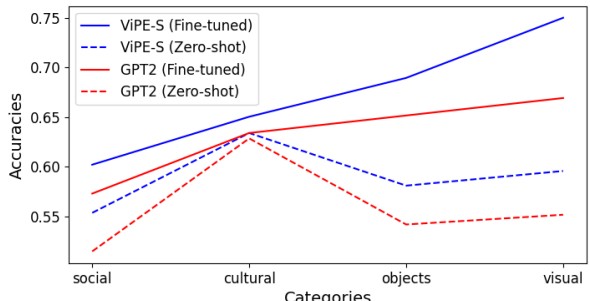

Figure 6: Profanity and offensive language analysis for lyrics with increasing profanity scores and those for visual elaborations generated by GPT2-M, ViPE-M, and GPT3.5. While GPT2-M's scores show a strong resemblance to that of pure lyrics, ViPE and GPT3.5 produce appropriate content across all intervals.

Figure 7: Zero-shot and fine-tuned evaluation results on different categories of the Fig-QA dataset (Liu et al., 2022b). ViPE-S outperforms GPT2 across all categories with a more pronounced gap in the visual category.

closely follow that of lyrics, indicating inappropriate language. GPT3.5 and ViPE on the other hand effectively reduce the profanity scores across all the intervals. These findings support ViPE's ability to transform inappropriate content and generate safe visual elaborations.

## 4.3 Human Evaluation

To strengthen our evaluation toolkit, we conducted a user study involving 30 native English-speaking participants aged between 20 and 40 for a comprehensive end-to-end assessment as follows:

**Data preparation:** From the HAIVMet dataset, we randomly selected 60 metaphors. For each metaphor, we generated visual elaborations using ChatGPT, ViPE, and added the human expert elaborations from HAIVMet. Subsequently, we employed Stable Diffusion to generate corresponding images from these visual elaborations.

**Experiment:** The experiment involved presenting participants with a metaphor alongside three images generated from prompts provided by human experts (HAIVMet dataset), ChatGPT, and ViPE. Their task was to choose the image that best represented the metaphor's meaning.

**Findings:** Our findings dovetail well with the previous results. Participants favored images from human experts 38.67% of the time, followed by ViPE's images at 33.61%, and ChatGPT's at 27.72%. These results validate ViPE's superiority over ChatGPT and its competitive performance with human experts.

## 5 Implementation details

**Training on LyricCanvas:** Two versions of ViPE are developed: ViPE-M (based on GPT2-small) and ViPE-S (based on GPT2-Medium). The models are fine-tuned on LyricCanvas for 5 epochs using 8 A100 Nvidia GPUs, each with 40 GB RAM. We use the AdamW optimizer (Loshchilov and Hutter, 2017) with a learning rate of $5e-05$ and a linear scheduler with 1000 warmup steps. For ViPE-S, the batch size is generally 50, except with a context size of 7, where a batch size of 32 is utilised. In the case of ViPE-M, the batch sizes vary for different context sizes: $\{32, 32, 20, 16, 8\}$ for context sizes $\{0, 1, 3, 5, 7\}$, respectively. 10 % of LyricCanvas ($\approx$ 1M samples) is used for validation.

**Image-text Retrieval:** We load a BLIP (Li et al., 2022a) checkpoint trained on COCO, initialised on ViT-B(Dosovitskiy et al., 2021) and BERT-base (Devlin et al., 2018). To finetune, we use a batch size of 16 for 10 epochs using AdamW, a learning rate of $1e - 4$, and a batch size of 128 with reranking for fast inference, commonly used in retrieval (Li et al., 2022a; Ghosh et al., 2023).

**Emotion Classification:** BERT-base-uncased is fine-tuned for 5 epochs using AdamW optimizer, a learning rate of $5e - 05$ and a batch size of 256.

**Figurative QA:** We made use of the provided evaluation framework[11] by Liu et al. (2022b) and trained with the batch size of 32 for 5 epochs using AdamW optimizer, with a learning rate of $5e - 05$. The test results are publicly available under the name *IMAGINATE EMNLP2023* on their leaderboard.

---

[11] https://github.com/nightingal3/Fig-QA

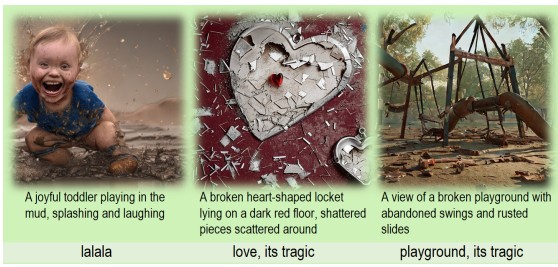

Figure 8: ViPE demonstrates robust contextual understanding across arbitrary textual inputs. Images are generated with ViPE elaborations and Stable Diffusion.

## 6 Applications

**Music Video Generation:** Besides its power to produce well-depictable prompts for text-to-image synthesis, we utilise ViPE as a versatile assistant to generate stylish visuals for music videos. We employ a zero-shot text-to-video synthesis approach, incorporating motion information for temporal consistency (Khachatryan et al., 2023) and smooth visual transitions between lines by interpolating image and text embeddings. More specifically, our approach comprises (1) extracting lyrics and timestamps from a given audio file, (2) generating visual elaborations from lyrics using ViPE, (3) creating a cohesive video narrative that encompasses the composition of the song, including all the lyrics and music. In Appendix B, we detail our pipeline. Below, we summarise our key points.

(1) ViPE generates relevant and visualisable elaborations. (2) To tackle the semantic coherence mismatch arising from diverse visual elaborations, we propose a novel approach called "Chunked Similarity Alignment." This technique aligns the most relevant parts of the latent representations during transitions. Overall, our finding further confirms the robustness of ViPE across various domains.

**Style Transfer and Creative Writing:** ViPE demonstrates robust contextual understanding across different domains. Figure 8 shows examples of images generated by Stable Diffusion using ViPE's elaborations. ViPE exhibits impressive generalization capabilities, even with non-lexical terms. More examples are available in Appendix C. These findings indicate that ViPE has applications in style transfer and creative text generation.

## 7 Conclusion

In this paper, we introduced ViPE, the first automated model for visualising figurative expressions in text-to-image models. ViPE efficiently gener-

ates diverse image captions, or visual elaborations, from arbitrary textual input. Our approach involves training lightweight language models on a novel dataset, LyricsCanvas, comprising 10 million lines of lyrics paired with visual elaborations generated by GPT3.5. Our key achievements are as follows: **(1)** We created the LyricsCanvas dataset, which enables training powerful language models for visualising figurative language. **(2)** We built ViPE by distilling the knowledge from GPT3.5 to a lightweight and open-source model with robust performance. ViPE exhibits highly robust zero-shot generation capabilities, surpassing GPT3.5 and achieving competitive results compared to human experts. **(3)** We demonstrated the versatility of ViPE by generating visually captivating elaborations from various textual inputs, such as non-lexical terms, abstract concepts, and figurative language. This opens up possibilities for applications in creative writing, paraphrase generation, and style transfer. **(4)** ViPE serves as a strong baseline for visualising lyrics, evident in the visually appealing artworks it generates for music video visualisations.

Overall, ViPE enables accessible solutions for complex research questions and paves the way for automated pipelines. In the future, we plan to apply ViPE in investigating the interplay between language and perception in related disciplines such as psycho-linguistics and cognitive science.

## Limitations

While we provide evidence of ViPE's robustness and understanding of figurative language using multiple benchmarks, the evaluation may have limitations. The choice of evaluation metrics and the specific datasets used for assessment may not fully capture the nuances and complexities of human figurative expressions. More specifically the cultural differences in creating and interpreting figurative phrases. Further investigation and comparative analysis with more diverse and perhaps new evaluation measures and data sets would strengthen the assessment of ViPE and potential future models.

## Ethics Statement

Even though The ViPE model has been fine-tuned on data generated by GPT3.5 with filtering which incorporates certain measures to mitigate biases, it is built upon the GPT-2 model. GPT2, despite its popularity, exhibits biases in job stereotypes, gender, and ethnicity distributions (Kirk et al., 2021).

Therefore, there is still a possibility that ViPE may exhibit similar biases. We believe it is crucial to remain vigilant in addressing and rectifying biases in language models. Moreover, it is important to note that popular text-to-image models might aggravate such biases, as they tend to over-represent aspects associated with whiteness and masculinity in their latent space (Luccioni et al., 2023).

## Acknowledgements

This work has been supported by EXC number 2064/1 – Project number 390727645, as well as by the German Federal Ministry of Education and Research (BMBF): Tübingen AI Center, FKZ: 01IS18039A. The authors thank the International Max Planck Research School for Intelligent Systems (IMPRS-IS) for supporting Hassan Shahmohammadi.

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

## A System Role

Below we provide the prompt, also known as System Role, that we used for instructing GPT3.5 Turbo to generate visual elaboration for a given set of lyrics.

Follow my commands: 1. Convert abstract lyrics into depictable prompts that represent the original lines, such as using "a man and a woman are having a conversation over a cup of tea" to represent "somebody once told me" and "a shining diamond ring" to represent "all that glitters is gold." 2. Keep prompts concise and avoid creating prompts longer than 20 words. 3. The requirement is to have one prompt per line. If there are 40 lines of input, the output should contain 40 prompts. 4. When generating prompts, do not focus on what the subject is thinking or feeling. For example, instead of "a student thinking about his long assignment list, overwhelmed by so much coursework," which is difficult to visualize, describe the student's appearance, such as "a male student looking at a long assignment list, with a scared expression, tears rolling down from his cheek." 5. Structure all prompts by setting a scene with at least one subject and a concrete action term, followed by a comma, and then describing the scene. For instance, "a view of a forest from a window in a cozy room, leaves are falling from the trees." 6. To add variety and avoid repetition, it is important to mix up singular and plural forms when referring to subjects or objects in the prompts. For example, "two cats," "ten men," "five girls," or "seven books" can be used instead of consistently using singular forms. 7. Some lyrics may contain inappropriate content, but the goal is to generate acceptable and decent prompts for them. 8. Consider the sentiment of the song when generating prompts. The same line should be represented differently depending on the mood of the song. For example, "I went for a walk" could be converted to "a young man is taking a walk on a sunny day in a beautiful park full of trees" if the song is positive, or "an old man is taking a walk at night in a dark forest full of trees" if the song is negative. 9. Do not use generic words such as person, people, man, woman, individual, figure, object, etc. Instead, across various topics, use diverse and specific terms such as desert, island, statue, skyscraper, stars, moon, rainbow, snowflakes, wolf, horse, dragon, bird, python, bike, truck, airplane, astronaut, daisies, roses, diamond ring, and so on, where appropriate. 10. Do not always use human subjects. For instance, instead of "A person standing under a starry night sky, aware that there is no tomorrow" use "A clock with its hands frozen, in a cold weather where everything is frozen". 11. Describe the scene with details and use various adjectives. For instance, colorful kites in the cloudy sky, frozen lakes with a gorgeous sunset in the background, a very long tree reaching the clouds, and so on.

For example, if I give you: "1. Feels like the weight of the world 2. Like God in heaven gave me a turn 3. Money could be dangerous 4. Everyone is leaving 5. This is gonna be the best day of my life 5. I am forever free"

I expect you to give me a prompt per line as follows:

"1. A man carrying a giant globe on his back in a post apocalyptic world, struggling with the weight. 2. A scary demon is spinning a wheel in the dark and gloomy sky. 3. A dragon with evil eyes is lying on a pile of shiny gold. 4. A picture of an abandoned city in dark gloomy weather, buildings are dark and destroyed. 5. A stunning fireworks display illuminating the night sky, people are happily dancing. 6. A majestic eagle soaring through the vast open sky, wings outstretched."

Prioritize Rules 9 and 10: don't use generic terms and human subjects while conveying the original lines. Start your response with "1.".

## B Music Video Generation: A Detailed Pipeline

In this section, we provide the detailed layout of the methodology used to convert song lyrics to music videos. In B.1, we delve into the steps taken to convert the song as an audio file into a suitable input for text-to-image Latent Diffusion Models, i.e. Stable Diffusion. In B.2, we detail the type of ViPE model used and the process of documenting the entire length of the song with visual elaborations. In B.3, we provide the steps taken in conducting zero-shot text-to-video generation for a given lyric and its corresponding visual elaboration, while in B.4, we talk about our interpolation mechanism and the strategy we use to mitigate semantic incoherence while interpolating in the text embedding space, as introduced in Section 6.

### B.1 Preprocessing

As the main inputs to the lyric-to-video pipeline, we only require the audio file, the title of the song,

and the artist's name to return a complete music video that illustrates the lyrics as they are being sung. From the audio file, we retrieve the song lyrics and an accurate set of timestamps $T_{N+1}$, that corresponds to lyrics $l_i$, such that there is a direct correspondence of $(T_{i+1}, T_i) \rightarrow l_i$, akin to fine-grained audio transcribing.

We do this by employing Whisper (Radford et al., 2022), a general-purpose speech recognition model which achieves state-of-the-art results by scaling the weakly supervised pre-training procedure, to provide the transcribed output of lyrics as well as its associated timestamps. Whisper performs well in understanding and splitting vocal sequences, at times clubbing two lines together if the gap between them is small. We argue that this is beneficial as sufficient time is given to visualise the lyrics in the video and we can manifest ViPE's context-learning capabilities too. The Whisper-large model, despite having 1550M parameters (compared to 769M for Whisper-medium) was chosen, as exact timestamps are required for the best possible audio-to-visual alignment.

## B.2 Visual Elaborations

We harness ViPE's ability to improve visual elaborations with an increased context size (Figure 8) in establishing a visual narrative in text form. In our experiments, we use ViPE-M, with a context size of 7, to sample Visual Elaborations of each line, which serves as the input to the Stable Diffusion. While it is established how well ViPE performs on Visual Elaborations, it also provides meaningful outputs to visualise the music break sections, where there are no lyrics to sample from.

For example, for Adele's Skyfall, we input *Skyfall by Adele; musical intro* and ViPE generates *Adele playing the piano in front of a large crowd*, which is easily visualisable. Towards the middle of the song, we can use the context of previous lines to establish a suitable prompt during musical breaks, such as *Pop band performing on the streets of New York in a thunderstorm* for Thunder by Imagine Dragons.

## B.3 Text-to-Video

With the availability of well-trained text-to-image Stable Diffusion Models, (Khachatryan et al., 2023), we established the procedure for zero-shot text-to-video generation. Our approach follows the Text2Video-Zero Diffusion pipeline (Khachatryan et al., 2023), which we tune to reduce the

occurrence of random motion between two frames. Implementing this algorithm gives us promising results, with a definite scope for improvement. As an alternative and additional experiment, we also implement the img2img Stable Diffusion[12] algorithm to generate images similar to an initial image fed to the model. We observe in Figure 10 that there are some variations based on the noise injected in the diffusion process, it is not varied and temporally consistent. We also see that Text2Video-Zero causes errors between two frames as shown in Figure 10.

## B.4 Interpolation

For video generation, besides using zero-shot text-to-video generation, we establish a novel embedding-chunking strategy in the text latent manifold for text-to-image generation to incrementally interpolate the latent representations, thus providing interesting visual transitions from one line to the next.

We use spherical linear interpolation (slerp) to interpolate between two subsequent CLIP text embeddings of shape (77,768) as well as two conditional latents which are denoised in the diffusion process. By smoothly interpolating between latents as highlighted in Equation 2 and recent works like (Bhunia et al., 2023), we maintain a visually appealing and coherent transition between two lines instead of cutting from one distinct scene to another.

$$slerp(p_0, p_1, t) = \frac{\sin(1-t)\theta}{\sin\theta} p_0 + \frac{\sin t\theta}{\sin\theta} p_1 \quad (2)$$

However, this interpolation formulation does not account for semantic coherence in the text latent space. Before interpolating between $p_0$ and $p_1$, we argue that it is essential that interpolated embeddings correspond to similar text. To reduce the risk of introducing non-coherent intermediate embeddings, we implement a sliding window approach to align subsets of the text embeddings of $p_1$ to reference $p_0$, which we call *Chunked Similarity Alignment*.

We rearrange $p_1$ by selecting the window with the highest cosine similarity (we capture the semantic relation between words based on the alignment of their semantic orientations). Our approach helps preserve semantic coherence and continuity during

---

[12]https://huggingface.co/spaces/fffiloni/stable-diffusion-img2img

interpolation, which is crucial considering the diversity of visual elaborations that need to be illustrated, as otherwise interpolating may lead to incoherent transitions between disparate texts. We showcase frame-to-frame results to compare traditional slerp interpolation and interpolation with Chunked Similarity Alignment, in Figure 9. We qualitatively show how aligning similar embeddings on the text manifold aids better interpolation on the image manifold.

## C   Creative Visual Elaborations

In this section, we present additional examples to demonstrate the extensive capabilities of ViPE in comprehending various non-literal expressions and producing credible visual elaborations accordingly. The results are shown in Figure 11 for Stable Diffusion and Figure 12 for DALL·E 2. While both models struggle to visualize complex textual inputs, ViPE excels in generating visually comprehensible elaborations while maintaining the semantic integrity of the arbitrary textual prompts.

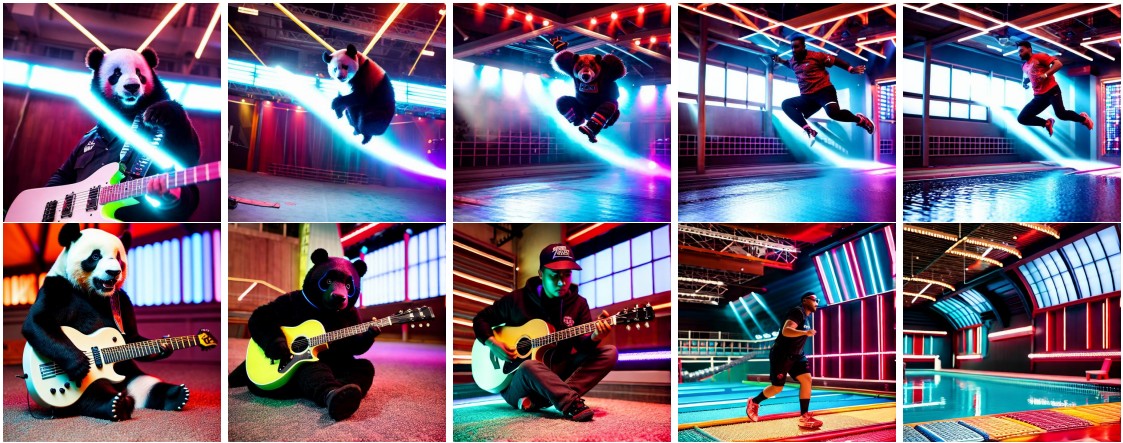

Figure 9: Comparison between interpolation methodologies: *Chunked Similarity Alignment*(Ours)(**Top**) and normal slerp interpolation(**Bottom**). We interpolate between A panda playing the guitar to A man jumps into a pool. Normal interpolation fails by being less smooth in transitions between frames, which is important during video generation as we normally use high fps rates (we use 10 in the music videos we generate) and such variations between frames would make the interpolation noisy as a sequence. Normal interpolation also fails to align transitions as well as our approach as can be shown in the transition rate between the panda transitioning to a man and also the floor transitioning to a pool.

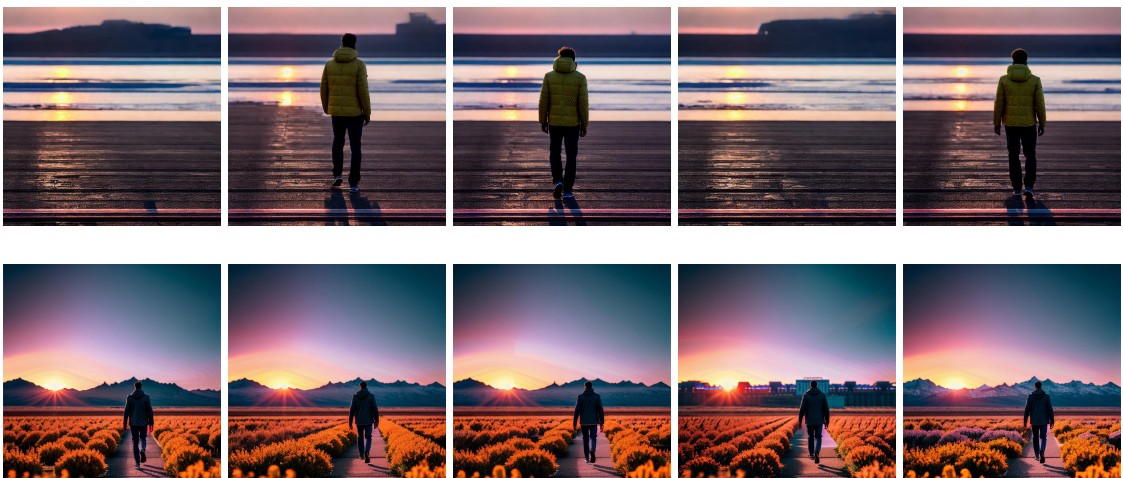

Figure 10: Comparison between frame-to-frame sequential image generation: **Text2Video-Zero**(Khachatryan et al., 2023)(Top) and img2img Stable Diffusion(bottom) generating images for the prompt Man walks towards a beautiful sunrise, looks into the distance, while visualising Adele's Skyfall. Both models have difficulty with temporal consistency and motion, while img2img has low variations between frames, Text2Video-Zero has high variations.

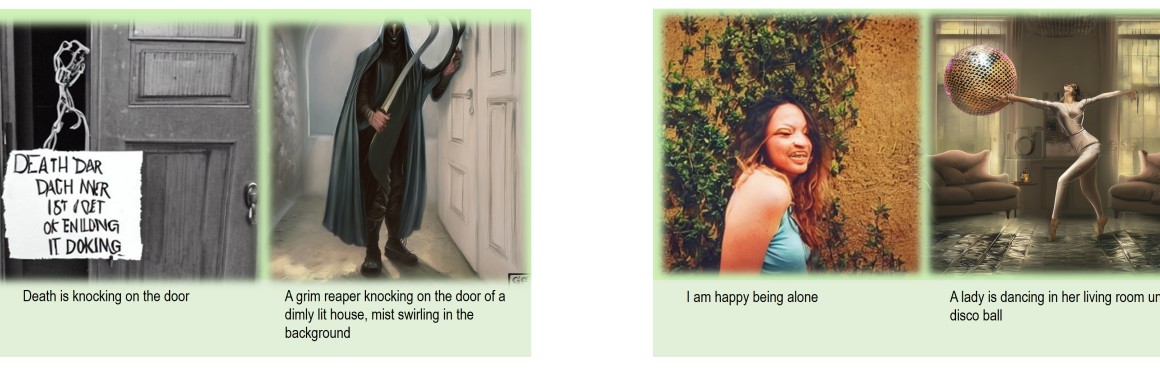

| Death is knocking on the door | A grim reaper knocking on the door of a dimly lit house, mist swirling in the background | I am happy being alone | A lady is dancing in her living room under a disco ball |

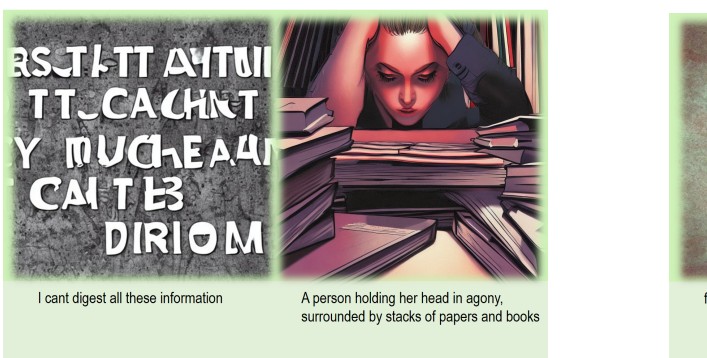

| I cant digest all these information | A person holding her head in agony, surrounded by stacks of papers and books |

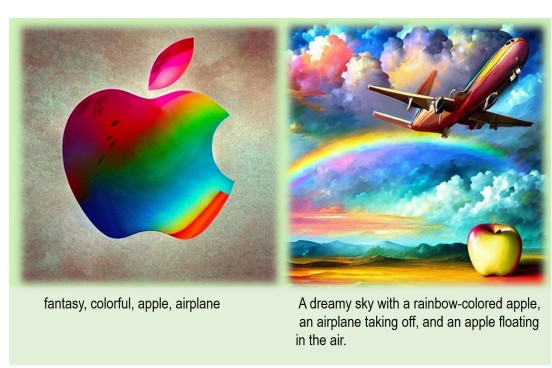

| fantasy, colorful, apple, airplane | A dreamy sky with a rainbow-colored apple, an airplane taking off, and an apple floating in the air. |

Figure 11: Qualitative evaluations using Stable Diffusion with and without ViPE's visual elaborations. The left column in each sub-figure displays the prompt and Stable Difussion's output, while the other columns show ViPE's interpretations and the resulting image.

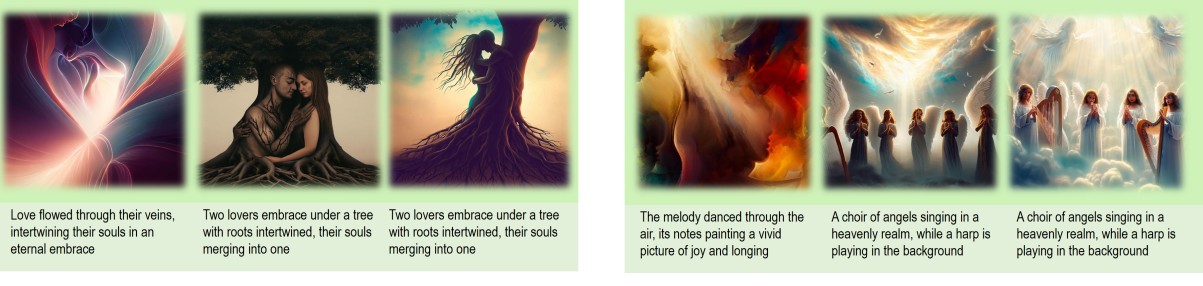

| Love flowed through their veins, intertwining their souls in an eternal embrace | Two lovers embrace under a tree with roots intertwined, their souls merging into one | Two lovers embrace under a tree with roots intertwined, their souls merging into one | The melody danced through the air, its notes painting a vivid picture of joy and longing | A choir of angels singing in a heavenly realm, while a harp is playing in the background | A choir of angels singing in a heavenly realm, while a harp is playing in the background |

Figure 12: Qualitative evaluations using DALL.E 2 with and without ViPE's visual elaborations. The left column in each sub-figure displays the prompt and the image generated by DALL.E-2, while the other columns show ViPE's interpretations and the resulting images.