# OpenReview forum: "ViPE: Visualise Pretty-much Everything"
_EMNLP/2023/Conference — EMNLP 2023 Main_

### Official Review · Reviewer_jo7U · 2023-08-02

**Soundness:** 3

**Excitement:**

4: Strong: This paper deepens the understanding of some phenomenon or lowers the barriers to an existing research direction.

**Paper Topic And Main Contributions:**

This paper presents a model ViPE capable of transforming metaphorical phrases or abstract concepts to visual elaborations which can be used to instruct DALLE.2 for a visualization.

The authors propose to collect a large-scale lyric dataset as a proxy of figurative language, then instruct ChatGPT to provide noisy visual elaborations given the lyrics. Finally, the lyric-vision elaboration pair distills a small model (GPT2-small & GPT2-medium).

Experimental results show that the trained models can generate better visual elaborations ( measured by the BLIP recall score) than GPT 3.5 and human experts. Further results on the tweet emotion classification dataset, show that the visual elaboration generated by ViPE results in higher classification accuracy and better alignments with the synthetic images.


**Reasons To Accept:**

- The visualize everything idea is exciting and of great potential.

- The released resources would be helpful for text-to-image applications and studies.



**Reasons To Reject:**

- The technical contribution, i.e., generating a dataset with ChatGPT and then conducting distillation, is weak.

- The evaluation results are mostly conducted implicitly. A  human evaluation on the triple of metaphor, visual elaboration and generated image quality is needed to fully convince the readers (small scale would be enough).

**Reproducibility:**

4: Could mostly reproduce the results, but there may be some variation because of sample variance or minor variations in their interpretation of the protocol or method.

**Reviewer Confidence:**

4: Quite sure. I tried to check the important points carefully. It's unlikely, though conceivable, that I missed something that should affect my ratings.

---

> ### Author Rebuttal · Authors · 2023-08-28
>
> Dear Reviewer,
>
> We sincerely appreciate the time and effort you invested in reviewing our work. Your insightful comments have contributed to the enhancement of our approach. Here, we address the raised points:
>
>
>
> ### The Technical Contribution
> As highlighted in our paper, "ViPE enables accessible solutions for complex research questions". We believe that simplicity and effectiveness should be prioritized over complexity when both yield equivalent outcomes. Given ViPE's clear advantages over ChatGPT in terms of computational efficiency (__175B vs 345M__ parameters) and performance for figurative language visualization, we view the simplicity of our approach as a notably positive and elegant solution.
>
>
> ### Direct Human Evaluation
> Thanks for pointing this out, in response, we have conducted a user study involving 30 native English-speaking participants aged between 20 and 40 for a comprehensive end-to-end assessment.
>
> - __Data preparation:__ From the HAIVMet dataset, we randomly selected 60 metaphors. For each metaphor, we generated visual elaborations using ChatGPT, ViPE, and added the human expert elaborations from HAIVMet. Subsequently, we employed Stable Diffusion to generate corresponding images from these visual elaborations.
>
> - __Experiment:__ The experiment involved presenting participants with a metaphor alongside three images generated from prompts provided by human experts (HAIVMet dataset), ChatGPT, and ViPE. Their task was to choose the image that best represented the metaphor's meaning.
>
> - __Findings:__ Our findings dovetail well with the results in our paper. Participants favored images from __human experts 38.67%__ of the time, followed by __ViPE's images at 33.61%__, and __ChatGPT's at 27.72%__. These results validate ViPE's superiority over ChatGPT and its competitive performance with human experts. We eagerly add these insights to our paper, inspiring further research to bridge the gap toward human-level expertise.
>
>
> Thank you again for your comments and suggestions, we hope our responses and new experiments have effectively addressed the concerns you raised.
>
> Best,
>
> The Authors

---

### Official Review · Reviewer_Wr1n · 2023-08-05

**Soundness:** 3

**Excitement:**

4: Strong: This paper deepens the understanding of some phenomenon or lowers the barriers to an existing research direction.

**Paper Topic And Main Contributions:**

This paper is about ViPE, a set of robust and lightweight language models that can generate visual elaborations from arbitrary text input, such as lyrics, tweets, or figurative expressions. Visual elaborations are textual descriptions of visual scenes that convey the intended meaning of the original text.

The paper addresses the problem of how to enhance the visualization of figurative and non-literal expressions in any text-to-image model. The main contributions that it makes towards a solution or answer are:

+ Introducing a dataset available for generating visual elaborations, called LyricCanvas, which contains about 10 million samples of lyrics and synthetic visual descriptions generated by a large language model (LLM).
+ Proposing a symbolic knowledge distillation method to train a robust and powerful model (ViPE) by fine-tuning lightweight language models on LyricCanvas.
+ Demonstrating the effectiveness and versatility of ViPE in various applications, such as image-text retrieval, emotion visualization, music video generation, and figurative language understanding.

**Questions For The Authors:**

A) How did you handle cases where the input text is not figurative or abstract, but literal or concrete? Did you generate visual elaborations for such cases as well, or did you filter them out or use a different approach?

B) How did you measure the semantic proximity and visual perceptibility of the generated visual elaborations? Did you use any metrics or tools to quantify these aspects, or did you rely on human judgment or intuition?

**Reasons To Accept:**

The strengths of this paper are:

+ It presents a novel and creative approach to generate visual elaborations from arbitrary text input, which can assist text-to-image models in visualising figurative and abstract phrases.
+ It introduces a large-scale dataset of lyrics and visual descriptions, which can be used for fine-tuning powerful language models or for other downstream tasks.
+ It evaluates the model on various benchmarks and applications, showing its superior performance and versatility compared to existing methods.

The main benefits to the NLP community if this paper were to be presented at the conference or accepted into Findings are:

+ It would inspire new research directions on figurative language visualisation, which is an under-explored but important topic.
+ It would provide a useful resource for researchers and practitioners who want to generate visual elaborations from text or enhance their text-to-image models.
+ It would showcase the potential of symbolic knowledge distillation as a general technique for transferring knowledge from large-scale noisy data to lightweight models.

**Reasons To Reject:**

Some possible weaknesses of this paper are:

+ It relies on an LLM to generate noisy visual descriptions, which may introduce errors or biases that affect the quality of the final output. For example, the LLM may generate inconsistent or irrelevant descriptions for some lyrics, or may fail to capture the nuances or emotions of the original text.
+ It does not conduct a human evaluation or user study to assess the subjective aspects of the generated visual elaborations, such as creativity, diversity, or emotional impact. It would be useful to collect feedback from human judges or users on how they perceive and appreciate the visual elaborations, and whether they find them helpful or enjoyable.

Some possible risks of having this paper presented at the conference or accepted into Findings are:

+ It may raise ethical concerns about the use of lyrics as a source of data, especially if they contain sensitive or offensive content. The paper does not discuss how it handles such cases, or whether it applies any filtering or moderation techniques to ensure the appropriateness and safety of the generated visual elaborations.
+ It may overlook some limitations or challenges of figurative language visualisation that are not addressed by the paper. For example, the paper does not consider the cultural or contextual differences in interpreting figurative expressions, or the ambiguity or uncertainty that may arise from generating multiple visual elaborations for the same text.

**Reproducibility:**

3: Could reproduce the results with some difficulty. The settings of parameters are underspecified or subjectively determined; the training/evaluation data are not widely available.

**Reviewer Confidence:**

3: Pretty sure, but there's a chance I missed something. Although I have a good feel for this area in general, I did not carefully check the paper's details, e.g., the math, experimental design, or novelty.

---

> ### Author Rebuttal · Authors · 2023-08-28
>
> Dear Reviewer,
>
> We sincerely appreciate the time and effort you invested in reviewing our work. Your insightful comments have contributed to the enhancement of our approach. Here, we address the raised points:
>
>
> ### Training on noisy data generated by LLM
> We agree that LLM including Chatgpt might generate noisy labels. However, incorporating synthetic noise has been demonstrated to enhance the robustness of machine translation, a task closely aligned with ours, as seen in studies such as [Improving Robustness of Machine Translation with Synthetic Noise](https://aclanthology.org/N19-1190) (Vaibhav et al., NAACL 2019) and [Toward Robust Neural Machine Translation for Noisy Input Sequences](https://aclanthology.org/2017.iwslt-1.13) (Sperber et al., IWSLT 2017). Consequently, by training on noisy synthetic data, we anticipate a similar enhancement of our model's robustness which we confirm with various evaluation methods including a new human evaluation mentioned below.
>
> ### Human Evaluation
> Thanks for pointing this out, in response, we have conducted a user study involving 30 native English-speaking participants aged between 20 and 40 for a comprehensive end-to-end assessment.
>
> - __Data preparation:__ From the HAIVMet dataset, we randomly selected 60 metaphors. For each metaphor, we generated visual elaborations using ChatGPT, ViPE, and added the human expert elaborations from HAIVMet. Subsequently, we employed Stable Diffusion to generate corresponding images from these visual elaborations.
>
> - __Experiment:__ The experiment involved presenting participants with a metaphor alongside three images generated from prompts provided by human experts (HAIVMet dataset), ChatGPT, and ViPE. Their task was to choose the image that best represented the metaphor's meaning.
>
> - __Findings:__ Our findings dovetail well with the results in our paper. Participants favored images from __human experts 38.67%__ of the time, followed by __ViPE's images at 33.61%__, and __ChatGPT's at 27.72%__. These results validate ViPE's superiority over ChatGPT and its competitive performance with human experts. We eagerly add these insights to our paper, inspiring further research to bridge the gap toward human-level expertise.
>
>
> ### Appropriateness and safety of the generated visual elaborations
> Language models such as GPT2 might indeed generate inappropriate content. We have therefore constructed safe training data by introducing an appropriateness criterion guiding the generation of visual elaborations within our training data, as also highlighted in our paper: "__Appropriateness:__ Some lyrics contain inappropriate content, so generated output should not explicitly describe such content". [__Footnote__: We automatically discarded those lyrics that were not processed by the system due to inappropriate content].
>
> - __New profanity and offensive language evaluation:__ To measure the appropriateness of ViPE's output, we conducted the following experiment. Using the [Alt-profanity-check library](https://pypi.org/project/alt-profanity-check/), we first measured the profanity score (inappropriate/offensive language) of the lyrics in the valuation set (around 1M line of lyrics). We then measured the profanity scores of the generated visual elaborations from ChatGPT and ViPE. In addition, we prompted the pre-trained GPT2 model with the lyrics and generated new text (not necessarily a visual elaboration). Subsequently, we measured the profanity score for the GPT2’s output.
>
> - __Results:__ Reported in the table Below, the mean and standard deviation of the lyrics's scores are (0.1778, 0.2615), while these numbers are still high for the pre-trained GPT2, we can see that __Both ViPE and ChatGPT results in scores close to zero__. In the paper, we will add a new figure showing how each of the three models behaves for inputs with increasing profanity scores.  For ViPE and ChatGPT, two closely positioned horizontal lines are observed, while GPT2’s plot shows a steady slope. Overall, these findings confirm that ViPE does not output inappropriate responses and closely matches ChatGPT's profanity scores.
>
> | Model   | Mean   | Standard Deviation  |
> |---------|--------|---------------------|
> | Lyrics  | 0.1778 | 0.2615              |
> | GPT2-M  | 0.1376 | 0.2315              |
> | ChatGPT | 0.0812 | 0.0968              |
> | ViPE-M  | 0.0962 | 0.102               |
>
>
>
> ### Cultural or contextual differences in interpreting figurative expressions
> Visualizing and interpreting figurative expressions is a very challenging task, primarily due to the substantial influence of subjectivity and the legitimacy of diverse interpretations. Moreover, the problem of underrepresenting nuanced cultural differences is apparent in many vision-language tasks comprising of mainly English datasets such as text to image synthesis. While our model may also exhibit such biases, its ability to generate multiple interpretations for a single input (e.g, a metaphor) is very promising. As an illustration, consider the range of plausible interpretations ViPE offers for the term _ambiguity_:
>
> 1. A person lost in a maze holding a map and looking confused
>
> 2. A bunch of tangled cables with similar colors
>
> 3. A complicated maze of mirrors, reflecting a distorted view of a woman’s face
>
> 4. A confused chameleon hiding behind a leaf on a sunny day.
>
> By generating various valid interpretations, ViPE significantly enhances its capability to encompass the spectrum of cultural and subjective differences inherent in such expressions.
>
>
>
> ### Questions
>
> ___Visual elaboration for concrete phrases:___ We did generate visual elaborations for concrete phrases with the same pipeline. While the generated elaboration in such cases conveys the main essence of the input, it typically encompasses more details about the scene. For instance, _'I went for a walk’_ might be converted into _'A man is walking in a park full of trees, the sun is shining’_.  By training on both concrete and figurative phrases, ViPE is able to construct visual elaborations from a wide range of keywords. For instance, for the input _‘fantasy, brave, apple’_, ViPE generates _`A brave knight holding a giant apple, standing in front of an enchanted castle'_
>
> __Semantic proximity and visual perceptibility:__
> - Semantic Proximity (SP) measures how well the generated visual elaboration v_i represents the meaning of the tweet t_i, determined by the final classification accuracy on D_v. In other words, once the emotionally charged tweets have been converted into visual elaborations, if the information about the emotion is well represented, then the classification score (using BERT) is higher.
>
> - Visual Perceptibility (VP) assesses the visualisability of the visual elaboration v_i by computing the cosine similarity between the CLIP embeddings of v_i and its corresponding generated image I_i by Stable Diffusion. In other words, if the generated visual elaboration is accurately visualizable, it results in a high-quality image and hence a high CLIP score with the visual elaboration.
>
> Thank you again for your comments and suggestions, we hope our responses and new experiments have effectively addressed the concerns you raised.
>
> Best,
>
> The Authors

---

### Official Review · Reviewer_eBD7 · 2023-08-10

**Soundness:** 5

**Excitement:**

5: Transformative: This paper is likely to change its subfield or computational linguistics broadly. It should be considered for a best paper award. This paper changes the current understanding of some phenomenon, shows a widely held practice to be erroneous in someway, enables a promising direction of research for a (broad or narrow) topic, or creates an exciting new technique.

**Paper Topic And Main Contributions:**

Topic:

This paper introduces ViPE (Visualise Pretty-much Everything), a novel approach for visualizing figurative expressions in text-to-image models. The paper addresses the challenge of synthesizing meaningful images for abstract and figurative expressions, a task with which current state-of-the-art models like DALL.E 2 and Stable Diffusion struggle. (Multimedia model, NLP and CV, Multimedia dataset)

Main Contributions:

1.	Introduction of ViPE: A collection of lightweight and robust language models trained on a large-scale set of lyrics with noisy visual descriptions. These models are capable of transforming any arbitrary text into a visualizable description.

2.	Creation of the LyricCanvas Dataset: A comprehensive dataset containing 10 million lines of lyrics paired with visual elaborations generated by GPT3.5. This dataset enables the training of powerful language models for visualizing figurative language.

3.	Robust Performance: Evidence demonstrating that ViPE is more robust than GPT3.5 in synthesizing visual elaborations. It also exhibits an understanding of figurative expressions comparable to human experts.

4.	Versatility and Applications: The paper showcases the potential of ViPE across various domains, including music video generation, style transfer, creative writing, and more.
5.	Open Source Availability: The source code, pre-trained ViPE models, and the LyricCanvas dataset are made publicly available.


**Reasons To Accept:**

1.	Innovation: The paper presents a novel approach to a challenging problem in NLP, offering a unique solution for visualizing abstract and figurative language.
2.	Rich Dataset Creation: The introduction of the LyricCanvas dataset, a significant contribution to the field, enables further research and development.
3.	Practical Applications: The paper outlines real-world applications of ViPE, making it relevant to both academia and industry.
4.	Comprehensive Evaluation: The paper provides compelling evidence of ViPE's effectiveness, robustness, and competitive performance compared to existing models.
5.	Open Source Contribution: The availability of the source code and dataset promotes transparency and encourages further research and collaboration. The attached publicized videos effectively present the novelty of the proposed model.


**Reasons To Reject:**

1.	Potential Bias Issues: The paper acknowledges that ViPE may exhibit biases similar to GPT2 in job stereotypes, gender, and ethnicity distributions. The mitigation of these biases is not thoroughly addressed.
2.	Evaluation Limitations: The choice of evaluation metrics and specific datasets may not fully capture the nuances of human figurative expressions. A more diverse and perhaps new evaluation measures and datasets might strengthen the assessment. Also more similar models should be meanwhile compared and evaluated to prove ViPE’s robustness.
3.	Lack of Detailed Analysis on Limitations: While the paper mentions limitations, a more in-depth analysis of potential pitfalls or challenges in implementing ViPE might have provided a more balanced view.
4.	Potential Over-Reliance on Existing Models: ViPE's reliance on GPT3.5 for generating visual elaborations might raise concerns about its independence and the potential need for further innovation in this area.


**Reproducibility:**

4: Could mostly reproduce the results, but there may be some variation because of sample variance or minor variations in their interpretation of the protocol or method.

**Reviewer Confidence:**

4: Quite sure. I tried to check the important points carefully. It's unlikely, though conceivable, that I missed something that should affect my ratings.

---

> ### Author Rebuttal · Authors · 2023-08-28
>
> Dear Reviewer,
>
> We sincerely appreciate the time and effort you invested in reviewing our work. Your insightful comments have contributed to the enhancement of our approach. Here, we address the raised points:
>
> ### Evaluation Limitations
> We do acknowledge that evaluating the task at hand is none trivial as also stated in the limitation section of our paper. To strengthen our evaluation toolkit, we have conducted a user study involving 30 native English-speaking participants aged between 20 and 40 for a comprehensive end-to-end assessment as follows:
>
> - __Data preparation:__ From the HAIVMet dataset, we randomly selected 60 metaphors. For each metaphor, we generated visual elaborations using ChatGPT, ViPE, and added the human expert elaborations from HAIVMet. Subsequently, we employed Stable Diffusion to generate corresponding images from these visual elaborations.
>
> - __Experiment:__ The experiment involved presenting participants with a metaphor alongside three images generated from prompts provided by human experts (HAIVMet dataset), ChatGPT, and ViPE. Their task was to choose the image that best represented the metaphor's meaning.
>
> - __Findings:__ Our findings dovetail well with the results in our paper. Participants favored images from __human experts 38.67%__ of the time, followed by __ViPE's images at 33.61%__, and __ChatGPT's at 27.72%__. These results validate ViPE's superiority over ChatGPT and its competitive performance with human experts. We eagerly add these insights to our paper, inspiring further research to bridge the gap toward human-level expertise.
>
>
> ### Reliance on GPT3.5
> We agree that the performance of ViPE is influenced by GPT3.5. However, we would like to emphasize that 1) this reliance on ChatGPT is mitigated by many benefits including the ones you have also laid out. 2) The rapid advancements observed in large language models, such as ChatGPT, are indeed promising. Our work stands out by being the first to showcase the robustness of symbolic knowledge distillation for figurative language visualization. This paves the way for future improvements of ViPE (e.g., fine-tuning on small humanly annotated data) by the research community.
>
>
> ### Potential Bias Issues
> The problem of underrepresenting nuanced cultural differences and over dominance of certain ethnicities and gender types are apparent in many vision-language tasks comprising mainly English datasets (e.g., text-to-image synthesis). While our model may also exhibit such biases, its ability to generate multiple interpretations for a single input (e.g., a metaphor) is very promising. As an illustration, consider the range of plausible interpretations ViPE offers for the term 'ambiguity':
>
> 1. a person lost in a maze holding a map and looking confused
>
> 2. a bunch of tangled cables with similar colors
>
> 3. a complicated maze of mirrors, reflecting a distorted view of a woman’s face
>
> 4. a confused chameleon hiding behind a leaf on a sunny day.
>
> By generating various valid interpretations, ViPE significantly enhances its capability to encompass the spectrum of cultural and subjective differences inherent in such expressions.
>
>
> Thank you again for your comments and suggestions, we hope our responses and new experiments have effectively addressed the concerns you raised.
>
> Best,
>
> The Authors

---

### Official Review · Reviewer_9QUY · 2023-08-13

**Soundness:** 3

**Excitement:**

4: Strong: This paper deepens the understanding of some phenomenon or lowers the barriers to an existing research direction.

**Paper Topic And Main Contributions:**

The paper introduces ViPE: Visualise Pretty-much Everything, a tool for generating visual descriptions of any arbitrary piece of text, including figurative and non-literal expressions. It eliminates the need for human annotations or images with metaphorical contents. Authors offers a series of lightweight and robust language models that have been trained on a large-scale set of lyrics with noisy visual descriptions that represent their implicit meaning. Authors also evaluate the proposed methods with both Intrinsic and extrinsic evaluation.


**Reasons To Accept:**

The authors introduce the first automated model for visualizing figurative expressions in text-to-image systems, which I believe is a significant advancement for the NLP community. Furthermore, they have released LyricsCanvas, a dataset comprising 10 million lines of lyrics paired with visual elaborations, marking another notable contribution.






**Reasons To Reject:**

1. The Visual Elaborations collected rely heavily on GPT3.5, which serves as the performance benchmark for the dataset. How does a model trained on this dataset differ from directly employing GPT3.5?
2. I remain unconvinced by the evaluation methods, particularly the extrinsic evaluation. Given that the objective is to enhance text-to-image generation, why not evaluate the images produced post-ViPE generation? An open-source text-to-image generation toolkit, like stable diffusion, could be utilized for this purpose.

**Reproducibility:**

5: Could easily reproduce the results.

**Reviewer Confidence:**

2: Willing to defend my evaluation, but it is fairly likely that I missed some details, didn't understand some central points, or can't be sure about the novelty of the work.

---

> ### Author Rebuttal · Authors · 2023-08-28
>
> Dear Reviewer,
>
> We sincerely appreciate the time and effort you invested in reviewing our work. Your insightful comments have contributed to the enhancement of our approach. Here, we address the raised points:
>
>
> ### Reliance on GPT3.5
> We acknowledge that the performance of ViPE is influenced by GPT3.5. However, we would like to emphasize that this reliance on ChatGPT is mitigated by many factors that contribute to the significance of ViPE:
>
> - Most importantly, our novel approach of symbolic knowledge distillation results in a more robust model compared to ChatGPT which enhances the reliability and applicability of ViPE.
>
> - ViPE effectively reduces the need for human labor. This becomes apparent when comparing the LyricCanvas dataset's substantial size (10M) with the current HAIVMet dataset (less than 7K), which relies heavily on human experts.
>
> - An essential point to note is ViPE's efficiency in terms of computational power and memory usage. ViPE's impressive efficiency for figurative language visualization is evident as it employs __507__ times fewer parameters than ChatGPT, showcasing its practical feasibility and relevance.
>
> - The rapid advancements observed in large language models, such as ChatGPT, are indeed promising. Our work stands out by being the first to showcase the robustness of symbolic knowledge distillation for figurative language visualization. This paves the way for future improvements of ViPE by the research community.
>
>
>
> ### Difference between using ViPE and ChatGPT
>
> - ViPE is more robust than ChatGPT, we showed that through various evaluation techniques.
>
> - ViPE is open-source, promoting collaboration, while ChatGPT is available exclusively through payment, limiting its accessibility to a broader audience.
>
> - ViPE demonstrates its practicality by being resource-efficient, with a modest size of just a few Gigabytes. This characteristic opens doors for researchers to explore and expand upon ViPE's capabilities. In contrast, the development of ChatGPT remains confined to a handful of major tech companies.
>
> ### Post-ViPE text-to-image Evaluation
> You are right, the objective is to enhance text-to-image generation. For the extrinsic evaluation, __we do indeed__ conduct __post-ViBE__ evaluation. For example, the retrieval evaluation involves the following steps:
>
> - For a given metaphor from the HAIVMet dataset, we generate a visual elaboration using ViPE, and ChatGPT
>
> - From the generated visual elaboration, we generate images using Stable Diffusion
>
> - For an __end-to-end evaluation__, we conduct metaphor-image retrieval. That is, given a metaphor, retrieve one of its corresponding images and vice versa. As reported in the first two rows of Table 2, this is a Post-ViPE evaluation as you have defined.
>
> - We also conduct a retrieval evaluation between the generated visual elaborations and the images, which are denoted as Caption_sz and Caption_ft in Table 2.
>
> Moreover, to strengthen our evaluation toolkit, we have conducted a __user study__ involving 30 native English-speaking participants aged between 20 and 40 for a comprehensive end-to-end assessment as follows:
>
> - __Data preparation:__ From the HAIVMet dataset, we randomly selected 60 metaphors. For each metaphor, we generated visual elaborations using ChatGPT, ViPE, and added the human expert elaborations from HAIVMet. Subsequently, we employed Stable Diffusion to generate corresponding images from these visual elaborations.
>
> - __Experiment:__ The experiment involved presenting participants with a metaphor alongside three images generated from prompts provided by human experts (HAIVMet dataset), ChatGPT, and ViPE. Their task was to choose the image that best represented the metaphor's meaning.
>
> - __Findings:__ Our findings dovetail well with the results in our paper. Participants favored images from __human experts 38.67%__ of the time, followed by __ViPE's images at 33.61%, and ChatGPT's at 27.72%__. These results validate ViPE's superiority over ChatGPT and its competitive performance with human experts. We eagerly add these insights to our paper, inspiring further research to bridge the gap toward human-level expertise.
>
> Thank you again for your comments and suggestions, we hope our responses and new experiments have effectively addressed the concerns you raised.
>
> Best,
>
> The Authors

---

### Meta-Review · Area_Chair_xyjn · 2023-09-18

**Recommendation:** 5

**Metareview:**

The authors introduce ViPE (Visualise Pretty much Everything), an innovative suite of text-to-image models that emphasize the depiction of figurative and non-literal expressions. Key outcomes of ViPE encompass the lyrics-based pretraining corpus named 'LyricCanvas' and an assortment of pretrained text-to-image foundational models.

There is a unanimous consensus among reviewers that ViPE is both exciting and original. It ventures into the visualization of figurative language, a domain that, though crucial, remains largely uncharted, and does so with a fresh methodology.

---

### Decision · Program_Chairs · 2023-10-07

**Decision:**

Accept-Main

**Comment:**

The authors introduce ViPE (Visualise Pretty much Everything), an innovative suite of text-to-image models that emphasize the depiction of figurative and non-literal expressions. Key outcomes of ViPE encompass the lyrics-based pretraining corpus named 'LyricCanvas' and an assortment of pretrained text-to-image foundational models.

There is a unanimous consensus among reviewers that ViPE is both exciting and original. It ventures into the visualization of figurative language, a domain that, though crucial, remains largely uncharted, and does so with a fresh methodology.